# What Affects Vocational Teachers’ Acceptance and Use of ICT in Teaching? A Large-Scale Survey of Higher Vocational College Teachers in China

**DOI:** 10.3390/bs13010077

**Published:** 2023-01-16

**Authors:** Chengming Yang, Rifa Guo, Yiran Cui

**Affiliations:** 1Graduate School of Education, Beijing Foreign Studies University, Beijing 100089, China; 2Institute of Education, Tsinghua University, Beijing 100084, China

**Keywords:** ICT integration, teachers’ internal expectancy, behavioral intention to use ICT, instruction behavior, higher vocational college

## Abstract

This study aimed to explore what factors affect teachers’ acceptance and instructional use of ICT in Chinese higher vocational colleges. Grounded in the modified UTAUT model, the current study investigated the direct and indirect effects of teachers’ performance expectancy, effort expectancy, external conditions, and behavioral intentions on using ICT in teaching. A total of 6087 teachers from 219 vocational colleges in 28 provinces in China participated in a large-scale survey. Structural equation modeling revealed that the teachers’ psychological perceptions (including performance expectancy, effort expectancy, and intention to use ICT technology) and the external support conditions (including professional development support, infrastructures, the climate of organizational reform and innovation, and teacher performance assessment mechanisms) significantly directly affect the use of ICT in Chinese higher vocational college teachers’ teaching practice. Moreover, this study confirmed the mediating role of teachers’ intention to use ICT in teaching in the relationship between teachers’ psychological perceptions and ICT instructional usage behavior. However, there were differences in the significance of these variables in the chain effect of teachers’ intention to use ICT. These findings expand our understanding of the factors influencing ICT use in teaching among VET teachers in China, thus providing practical implications for higher vocational college managers to promote teachers’ ICT teaching behaviors.

## 1. Introduction

The ubiquity and intensive use of information and communication technology (ICT) has fundamentally affected how we work, the skills required by the labor market, and how we teach and learn. As an integral approach to empowering people for future employment and decent livelihoods, it has been argued that vocational education and training (VET) should be reformed and optimized accordingly with the development of ICT to remain relevant to the changing world of work [1]. The Education 2030 Agenda has recognized new possibilities enabled by ICT to VET. These include authentic learning experiences facilitated by virtual reality and simulation and improved teaching methods and pedagogical techniques [2]. Notably, there is a growing trend toward using ICT in VET reforms and innovations [3]. Promoting ICT’s effective use is an essential strategy for modernizing China’s VET [4]. For example, the Ministry of Education of China has published the Guidance on Further Promoting the Vocational Education Development of Informatization [5], which highlighted the importance of ICT for VET’s quality assurance and pointed out that vocational colleges should use ICT to innovate teaching methods and continuously improve teachers’ digital literacy.

Although many factors influence the successful use of ICT in VET, teachers’ attitudes and capabilities to use ICT in teaching are the most fundamental [6,7]. However, an interesting finding is that Chinese VET teachers’ attitudes and intentions toward the pedagogical use of ICT were polarized [8]. On the one hand, some teachers are actively concerned about the impact of new educational technology on VET teaching. They continuously improve their ICT skills through teacher training, open educational resources, and online learning communities to optimize instructional design and practice, enrich teaching methods and innovate teaching assessment methods [9]. On the other hand, some teachers believe that integrating ICT into teaching adds extra workload, increases the complexity of teaching, and introduces new uncertainties [10].

What accounts for this polarization? Researchers have explored possible factors influencing the effective integration of VET teaching and ICT, such as VET policy, teachers’ ICT literacy, ICT infrastructure, digital educational resources, and teacher training [11,12]. Given the centralized top-down education management system, previous studies have found that Chinese teachers are positioned in a mandatory professional environment in which they are just implementing policies and procedures developed by policymakers and administrators [13,14]. In Addition, teachers’ psychological perceptions, such as cognition, attitudes, and motivation, also have important implications for boosting the use of ICT in VET teaching [15]. However, few studies have simultaneously considered teachers’ psychological perceptions and external support conditions when investigating the mechanism of effective integration between VET teaching and ICT in the Chinese context.

The Unified Theory of Acceptance and Use of Technology [16] has been hailed as one of the most practical models for explaining and understanding the psychological perceptions, external support conditions, behavioral intentions, and usage patterns of users accepting and using new technology [17]. Therefore, this study attempts to reveal how ICT acceptance and use influence the teaching behavior of VET teachers based on the UTAUT model in the Chinese context, which has the largest VET scale worldwide.

## 2. Theoretical Background and Hypothesized Model for Current Study

### 2.1. Instructional Behavior with ICT of Vocational Education Teacher

Instructional behavior is a significant factor in ensuring teaching quality and reflects the teaching styles [18]. It refers to the collection of behaviors teachers use in different teaching contexts and sessions to achieve desired teaching objectives [19]. As a form of education more closely related to the industrial economy, VET needs to update teaching content and resources according to work content, adjust teaching methods according to the working process, and optimize the teaching environment according to the work environment [20]. Furthermore, to equip individuals with digital skills and competencies to adapt to the changing world of work [1], VET teachers need to further digitize their teaching methods and behaviors [21]. Therefore, the teaching behavior of VET teachers in this study refers to the behavior collection of their design, implementation, monitoring, and evaluation of teaching to facilitate students’ perception of the real world of work and to train the skills required by digitized work in a blended learning environment that integrates the physical and digital environments. 

Most previous studies focused on internal or external factors influencing changes in the teaching behavior of VET teachers. For instance, some researchers explored the effects of external factors such as organizational infrastructure, school progress in digital transformation, supportive conditions for teacher professional development, and curriculum support for using digital technologies in teaching by VET teachers [22]. Relevant research revealed the significant indirect and direct effects of Chinese vocational college teachers’ psychological perceptions (e.g., perception of usefulness, self-efficacy, perceived ease of use, gratification, and intention to use) on actual ICT use by studying 573 teachers from three public vocational colleges in two Chinese provinces [23]. However, few studies consider both teachers’ psychological perceptions and the external support environment when uncovering the mechanism and influencing factors for VET teachers’ behavior changes, especially in the Chinese context. 

In the general education field, some scholars have begun to look simultaneously at external and internal factors when exploring the factors influencing teachers’ instructional behavior with ICT support [24]. It was supported by previous research showing that secondary school teachers’ ICT self-efficacy, school management, and collegial collaboration positively influenced teachers’ ICT pedagogical use [25]. Therefore, it is necessary to further research the factors influencing the changes in VET teachers’ instructional behavior with an integrative perspective.

### 2.2. The Unified Theory of Acceptance and Use of Technology (UTAUT)

Based on a systematic analysis and comparison of the eight models related to technology acceptance and usage, including the TAM model, the TRA, the TPB, the C-TAM-TPB, the MPCU, the IDT, the MM, and the SCT [26], Venkatesh et al. [16] proposed the UTAUT model. The original UTAUT model included six main constructs: performance expectancy, effort expectancy, facilitating conditions, social influence, behavioral intention to use technology, and usage behavior. Gupta et al. used the UTAUT to study the influencing factors of ICT usage in government organizations in a developing country [27]. Smirani and Boulahia investigated the application of open educational resources (OER) based on the UTAUT and found that performance expectancy and expected effort positively affect faculty members’ intention to use OER [28]. Moreover, it has been verified in the use of ICT and the context of China. For instance, Deng et al. used the modified UTAUT model to study the decisive factors of web-based question-answer service adoption at a Chinese university [29]. The above studies that validated or extended UTAUT in revealing teachers’ acceptance and use of technology primarily focused on elementary and higher education, but few researchers paid attention to VET teachers. Therefore, the UTAUT is employed in the current study to explore the VET teachers’ ICT acceptance and use in teaching.

### 2.3. Factors Influencing Teachers’ ICT Instructional Behavior Based on the UTAUT Model

#### 2.3.1. Performance Expectancy

Performance expectancy is the extent to which a user believes using technology will benefit and enhance their performance in certain activities [30]. It can be analyzed and measured by perceived usefulness [31]. Chiu and Wang noted that performance expectancy has a positive effect on people’s behavioral intention to use new technology, which also promotes the occurrence of users’ specified behavior [32]. As the main body of classroom activities, teachers and students are the most frequent users of technology in the teaching process. It has been explored that performance expectancy, which indicates the students’ perceived extent of using mobile learning, will increase their academic performance [33]. It has also been proved that teachers’ performance expectancy of integrating new technologies into teaching will enhance students’ performance. Therefore, performance expectancy in this study reveals the degree to which teachers believe using ICT will benefit them in optimizing the teaching practice and improving learning outcomes. 

#### 2.3.2. Effort Expectancy

Effort expectancy can be prescribed as the users’ expectations of ease related to using technology, which includes the perceived degree of ease and complexity of use [16,29]. This study describes effort expectancy as teachers’ perceptions of how easy-to-use ICT is in Chinese higher vocational colleges. Numerous studies have suggested a substantial relationship between effort expectancy and users’ intentions to use new technologies. For example, Lee confirmed such a relationship when exploring users’ continuous intention to engage in e-learning activities [34]. Adetimirin also noted that perceived ease of use is one of the prerequisites for undergraduates to use ICT in their learning process [35].

#### 2.3.3. External Conditions

According to the original UTAUT model, the external determinants, which influence users’ acceptance and use of technology, are explained by social influence and facilitating conditions. The former elucidated “the degree to which an individual perceives those important others believe he or she should use the new system” [16]. The latter referred to users’ expectations of the support and resources available regarding technology usage [36]. Although the UTAUT model has gained widespread acceptance, there are concerns about its capacity to explain the influencing factors of individuals’ acceptance and use of technology. Hence, it has been modified or expanded. For example, some scholars suggested that perceived risk, satisfaction, attitude, and trust should be considered to complement the original UTAUT model [37]. Other external factors have also been pointed out to influence teachers’ use of new technologies in teaching, such as social networks, communication or exchanging information with peers, social media [38], organizational commitment [39], and teacher professional title promotion system [40]. Moreover, Zhang and Liu stressed that the impact of school reform and innovation policies, incentive measures, and teacher performance assessment mechanisms should be considered when analyzing the factors influencing teachers’ application of new teaching methods and tools for professional development in a Chinese context [14].

Therefore, external conditions in the current study include not only social influence and facilitating conditions proposed in the original UTAUT model but also other extra factors, such as school policies on reform and innovation, the climate of communication among colleagues about technology use, the faculty professional title promotion system, and teacher performance assessment mechanisms.

#### 2.3.4. Behavioral Intention to Use

Previous research also argued that the rise in technology use was caused by the teacher’s behavioral intention to use it [25]. Behavioral intention to use mainly describes the teacher’s expectation to carry out plans and decide on the use of ICT [41]. According to Ahmad, performance expectancy, effort expectancy, facilitating conditions, and social influence all directly and positively affect people’s use intention of technology [31]. Joo et al. found that the intention to use will lead to technology acceptance and usage behavior with the support of the above four aspects [42]. Therefore, the proposed model of this study will also examine the influence of the Chinese VET teachers’ teaching behavioral intentions to utilize ICT on their actual usage behaviors.

### 2.4. Hypothesized Model for the Current Study 

In this study, we used the modified UTAUT model to examine what and how the dimensions influence Chinese higher vocational college teachers’ using ICT in teaching. Compared with the original UTAUT model, we had the following changes: firstly, we combined both social influence and facilitating conditions as external conditions considering that both reveal external support. Secondly, we assumed that the above-mentioned factors affect teachers’ instructional behavior directly and indirectly through teachers’ intention to use ICT. Additionally, gender, working experience, and school level were considered as the control variables because they were the significant moderators on the decision of ICT usage summarized by the prior researchers [31,39,43]. Therefore, Figure 1 depicts the initially hypothesized mechanism of the Chinese higher vocational college teachers’ acceptance and usage of ICT in teaching.

When controlling for teachers’ gender, working experience, and school level, the following hypotheses were proposed:

**H1.** 
*Chinese higher vocational college teachers’ performance expectancy will positively and directly affect their actual instruction behaviors of ICT usage (path a2 in Figure 1).*


**H2.** 
*Chinese higher vocational college teachers’ performance expectancy will positively and directly affect their actual instruction behaviors of ICT usage through their intentions to use ICT in teaching (path a1*d in Figure 1).*


**H3.** 
*Chinese higher vocational college teachers’ effort expectancy will positively and directly affect their actual instruction behaviors of ICT usage (path b2 in Figure 1).*


**H4.** 
*Chinese higher vocational college teachers’ effort expectancy will positively and indirectly affect their actual instruction behaviors of ICT usage through their intentions to use ICT in teaching (path b1*d in Figure 1).*


**H5.** 
*External conditions will positively and directly affect Chinese higher vocational college teachers’ actual instruction behaviors of ICT usage (path c2 in Figure 1).*


**H6.** 
*External conditions will positively and indirectly affect Chinese higher vocational college teachers’ actual instruction behaviors of ICT usage through their intention to use ICT in teaching (path c1*d in Figure 1).*


## 3. Method

### 3.1. Participants

The current study surveyed 6087 teachers from 219 vocational colleges in 28 provinces of China, covering 82.4% of provincial administrative regions in China: (1) we randomly selected vocational colleges in each province; (2) we contacted the vocational college principals to introduce our research intentions and ask for their approval; (3) we sent questionnaires to faculty at the vocational colleges that agreed to participate. Written informed consent was indicated on the questionnaire, and the teachers participated entirely voluntarily and anonymously. Moreover, no outliers or missing values above 10% were removed by data screening.

All teachers were full-time certified teachers, of which 35.6% were from regular colleges, 29.4% were from key provincial colleges, and 35.0% were from key national colleges. Table 1 presents teachers’ personal information, including gender, age, years of teaching experience, years of working experience, professional rank, and their highest degree.

### 3.2. Measurements

#### 3.2.1. Predictive Variables

**Performance Expectancy.** It was measured by teachers’ perception of the advantages that arise from using ICT [44,45]. Seventeen items with a five-point Likert scale were used, where 1 means “strongly disagree” and 5 means “strongly agree”. It included two dimensions: Perceived usefulness for improving students’ learning results (named “Perceived usefulness for learning”, 11 questions, e.g., “It is more helpful to enhance students’ enthusiasm to participate in learning”), and Perceived usefulness for promoting teachers’ instructional practice (named “Perceived usefulness for instruction”, six questions, e.g., “It helps to visualize abstract knowledge”). Cronbach’s alpha reliability coefficients and the Confirmatory factor analysis (CFA) results of both sub-scales are shown in Table 2, which indicates both high reliability and construct validity.

**Effort Expectancy.** It assessed teachers on their ease of using various ICT tools in daily teaching [44,46]. Eight items with a five-point Likert scale were used, where 1 means “strongly disagree” and 5 means “strongly agree”. The example item was “I am familiar with the use of software to create digital teaching resources.” Higher scores indicate that teachers perceived it easier to use the relevant ICT tools. Cronbach’s alpha reliability coefficient was 0.920, and all component loadings ranged from 0.444 to 0.956, indicating high reliability and construct validity. Detailed information is shown in Table 2.

**External Conditions.** It was measured by the frequency with which teachers received supportive actions from institutions to promote ICT instruction, for example, “Provide training for teachers to promote their ICT teaching ability”, and “Set up ICT teaching reform projects”. Eight items with a five-point Likert scale were used, where 1 means “never or rarely” and 5 means “usually”. Cronbach’s alpha reliability coefficient was 0.940, and all component loadings ranged from 0.691 to 0.912, indicating high reliability and construct validity. Detailed information is shown in Table 2.

**Behavioral Intention to Use.** It was used as a mediator variable to measure the degree of teachers’ intention to apply ICT to teaching. Teachers rated their likelihood of using ICT in teaching on a scale of 1–5, with higher scores indicating a stronger behavioral intention to use ICT in teaching.

#### 3.2.2. Outcome Variable

**Teachers’ Instructional Behavior.** It was measured by the frequency with which various information technology applications are used in formal instruction. Thirteen items with a five-point Likert scale were used, where one means “never or rarely” and five means “usually”. Examples include “Support pre-class preview or after-class reflection based on the platform” and “Provide structure based on the platform to support students to carry out autonomous inquiry learning”. Cronbach’s alpha reliability coefficient was 0.936, and all component loadings ranged from 0.418 to 0.862, indicating high reliability and construct validity. Detailed information is shown in Table 2.

#### 3.2.3. Control Variables

**Gender**. Two points are scored: 1 indicates male, and 2 indicates female.

**Years of Teaching**. It refers to how many years the teacher spent in full-time teaching work.

**School Level.** It refers to the level of the institution where the teacher is located. Options 1, 2, and 3 represent general higher vocational colleges, the higher vocational college selected by *Regional High-level Higher Vocational College Development Project*, and the higher vocational college selected by *National High-level Higher Vocational College Development Project*. Therefore, the higher the choice value, the higher the school levels.

### 3.3. Data Analyses

Descriptive statistics and correlation analysis of variables were computed using IBM SPSS Statistics 22.0. Structure Equation Modeling (SEM), tested by Mplus 7.4, was used to examine whether the hypothesized model fits the actual data. Good model fit results require the following indexes [47]: the root-mean-square error of approximation (RMSEA) should be lower than 0.08, and the comparative fit index (CFI) and Tucker–Lewis index (TLI) should be greater than 0.90. Moreover, the Sobel method was used to determine the effect size of the mediating effects, and the coverage of the total effect by indirect effects was calculated.

## 4. Results

### 4.1. Descriptive Statistics

The results of convergent, discriminant validity are demonstrated in Table 3. All the values of CR and AVE were more prominent than 0.50, which was the recommended cutoff score. Pearson Bivariate Correlation Analysis of variables was conducted on the mean values of each variable, and all variables have positive and significant relationships with each other. Thus, it allowed for further analysis of structural equation modeling.

### 4.2. Results of the Mediating Model

As shown in Figure 2, the results showed that the proposed model fits reasonably well with the data, namely, χ^2^/df = 23.05, RMSEA = 0.060, CFI = 0.905, TLI = 0.900, and all factor loadings were above 0.4.

As hypothesized, when controlling for gender, years of teaching, and school level, the paths directly connecting all the predictive variables and teachers’ instructional behavior were significantly positive. It can be inferred that if teachers perceived ICT to be more useful for improving students’ learning results (β = 0.330, *p* < 0.001), more useful for promoting teachers’ instructional practice (β = 0.183, *p* < 0.001), easier to use (β = 0.102, *p* < 0.001), and obtained more external support from institutions (β = 0.098, *p* < 0.001), they tended to use more ICT in their daily instructional behavior.

In addition to direct associations, the mediating role of teachers’ attitudes toward technology adoption was explored. Table 4 showed the coverage of indirect impacts on the total impacts. The chained mediating effect of “Perceived usefulness for learning” (effect size = 0.016, *p* < 0.001), “Perceived usefulness for instruction” (effect size = 0.009, *p* <0.001), “Effort expectancy” (effect size = 0.023, *p* < 0.001) and their “Instructional behavior” via connections through “Behavioral intention to use” were significant for the sample. Although the direct path between “External conditions” and “Teachers’ instructional behavior” was significant (β = 0.098, *p* < 0.001), the indirect effect of them influencing their attitude towards technology adoption was not significant (effect size = 0.003, *p* = 0.144).

## 5. Discussion 

Compared with previous studies, most of which only confirmed the respective effects of teachers’ psychological perception of ICT use [23] or incentive behavior from institutions [22], this study focused on the direct and indirect effects of the above factors on ICT application in teaching by verifying the modified UTAUT model. As hypothesized, the result indicated that teachers’ psychological perception and the external support conditions positively affect the pedagogical use of ICT in Chinese higher vocational college teachers’ teaching practice when controlling for gender, years of teaching, and school level. However, there were differences in the significance of these variables in the chain effect through teachers’ intention to use ICT: hypotheses H1, H2, H3, H4, and H5 were accepted, while hypothesis H6 was rejected.

### 5.1. What Directly Affects Teachers’ ICT Teaching Behavior?

Consistent with the hypotheses, the four predictive variables, including performance expectancy, effort expectancy, external conditions, and behavioral intention to use, were significantly related to Chinese higher vocational college teachers’ actual ICT use behavior in teaching. These positive findings can be explained by Bandura’s social learning theory, which emphasizes the interaction among environment, cognition, and behavior in social learning [48]. They also echoed previous studies on Chinese higher vocational college teachers’ ICT use behavior in teaching [23,49]. 

On the one hand, the findings suggested that Chinese higher vocational college teachers’ actual use of ICT in teaching was directly influenced by the college policies on reform and innovation, the climate of communication among colleagues about technology use, the faculty professional title promotion system, and teacher performance assessment mechanisms. The current results supported some similar conclusions of previous studies from the perspective of Chinese data [14,40]. For instance, Deng et al. found that people’s actual use of ICT tends to be a natural and voluntary behavior when they perceived that the external support conditions were available and accessible to them [29]. On the other hand, Chinese higher vocational college teachers’ actual use of ICT in teaching was directly affected by teachers’ psychological perceptions, such as performance expectancy, effort expectancy, and intention to use ICT in teaching [50,51]. This result is congruent with the UTAUT model [16]. It means that the higher teachers’ intention to use ICT, the more likely they will use ICT in teaching. Therefore, it is essential to pay attention to both the intrinsic traits of teachers and the external support conditions simultaneously to promote the pedagogical use of ICT by teachers in vocational colleges and improve teaching quality.

### 5.2. The Chained Mediating Effects of Teachers’ Intention to Use ICTs

The mediating role of teachers’ intention to use ICT in teaching in the relationship between teachers’ psychological perceptions and their instructional ICT behavior was supported in the current research, which is the same as previous research results [33]. For example, Tosuntaş et al. noted that teachers’ performance expectations and effort expectancy affected their actual use of an interactive whiteboard in teaching by influencing their behavioral intention.

Previous studies indicated that external conditions, such as social influence and facilitating conditions, positively affect users’ intention to use technology [52]. For instance, Al-Rahmi et al. proposed that social influence and facilitating conditions had positive effects on Malaysian university students’ intention to use social media to exchange and discuss information with peers [38]. However, our findings did not validate these results for Chinese higher vocational college teachers. 

One reasonable explanation for such a difference is that the long tradition of a centralized education management system and the solid managerial professionalism in the Chinese education field positioned teachers as deliverers of education reform policy [14,53,54]. Therefore, it is easy to know that the reform and changes in teachers’ teaching methods and behaviors tend to be mandatory and obligatory behavior. However, Venkatesh et al. proposed that social influence only works positively in a voluntary context [16]. Another explanation is that the mismatch between the needs of teachers’ professional development support and the support conditions provided by the school, such as training programs, digital education resources, and digital equipment, influences the increase in teachers’ intention to use ICT in teaching. According to the Vocational Education Informatization Development Report of China 2021 [55], the poor quality of digital education resources and the low matching of training programs for teachers’ professional development demands perceived by the Chinese vocational college teachers were the main influencing factors for the deep integration of ICT and teaching. These results also suggest that teachers’ intention to use ICT in teaching largely depends on their psychological perception in terms of willingness, attitude, and perception of usefulness and ease of use. 

An interesting finding was that the utility of the external support of the institutional environment was lower than the three variables related to teachers’ internal psychological perception, even the indirect effect through teachers’ intention to use was not significant. Therefore, it suggests that researchers and practitioners should pay special attention to teachers’ awareness of the usefulness and ease of use of information technology when promoting teachers’ application of ICT. Especially in the context of Chinese education that adapts to the top-down school management system [14], it is suggested that college managers should not only provide external support (such as teacher development training and policies) but also pay attention to teachers’ willingness to use, thus providing relevant support and services. Meanwhile, our findings revealed that Chinese higher vocational colleges need to consider the diverse and hierarchical needs of teachers’ pedagogical use of ICT when they design and develop external support conditions (e.g., training programs, digital education resources, and incentive mechanisms).

## 6. Conclusions and Implications

Based on the analysis of a large number of previous studies in the literature, this study uses the UTAUT model to construct the explanation model of influencing factors of vocational teachers’ acceptance and use of ICT to their teaching behavior from the perspective of teachers’ perceptions. A large-scale survey was conducted in 219 higher vocational colleges in 28 provinces of China and covered 6087 teachers. The influencing factors of higher vocational college teachers’ acceptance and use of ICT in their teaching behavior were analyzed. The results revealed that the higher vocational college teachers’ psychological perceptions and the external support conditions significantly directly affect the use of ICT in Chinese higher vocational college teachers’ teaching practice. Moreover, this study confirmed the mediating role of teachers’ intention to use ICT in teaching in the relationship between teachers’ psychological perceptions and ICT instructional usage behavior. This study employed and modified the UTAUT model in the Chinese context and confirmed the significance of simultaneously focusing on external support conditions and intrinsic teacher factors (e.g., expectancy, perception, attitude, and behavioral intention) in achieving high-quality development of ICT-enabled vocational education teaching.

### 6.1. Practical Implications

In the context of the implementation of vocational education digital transformation strategy in China, improving the digital awareness, literacy, and competence of vocational education teachers is an important assurance for improving teaching quality and innovative teaching methods. Therefore, this study puts forward the flowing suggestions for further promoting teachers’ effective and innovative use of ICT in Chinese higher vocational colleges:

Firstly, promoting teachers’ role in using new technologies to innovate teaching transformed from passive implementers into active explorers by empowerment. Influenced by the managerial professionalism in Chinese colleges, teachers are positioned in a mandatory professional environment in which they are just passive implementers of the education reform policies and practices developed by policymakers and administrators [14]. As a result, it tends to cause teachers’ rebellious psychology toward the pedagogical use of ICT and leads to a lack of internal motivation and positive attitudes. Therefore, to integrate ICT and teachers’ teaching in Chinese higher colleges, the significant role of teachers in school reform and pedagogical innovation should be clarified at the conceptual level. Meanwhile, at the practical level, teachers should be empowered to formulate teaching reform plans and provided an inclusive and diverse environment, to build a new professional development model of shared, collaborative, and autonomous.

Secondly, given the positive and direct effects of teachers’ perception of ease to use and usefulness on their intention and behavior to use ICT in teaching, we need to help teachers understand ICT’s potential for improving their teaching quality and how to use ICT more efficiently and effectively [56]. On the one hand, we need to explore typical examples of teachers where ICT has been used proactively and innovatively in teaching and achieved excellent results. It will help teachers feel the real effects of ICT-enabled teaching and increase their willingness to use ICT in teaching. On the other hand, it needs to ensure that teachers access high-quality, editable, and easy-to-use digital educational resources and tools. Therefore, school administrators need to create a more personalized and supportive environment.

Thirdly, considering the positive and direct effects of external conditions on teachers’ actual ICT usage behavior, the training mechanism for teachers should be reformed to improve teachers’ competence to use ICT efficiently and effectively in teaching. The continued enhancement of teachers’ digital literacy and competencies will help students develop the skills and competencies required by the digital world of work. Due to the diverse sources of teachers in higher vocational colleges and the various needs of teachers in different professional development phases, the principles refer to demand-oriented, classification, personalization, and dynamic optimization should be followed in the design of teacher training plans and programs.

### 6.2. Limitation and Future Work

Several limitations need to be considered. First, although we proved that teachers’ internal psychological perception and external factors supported by institutions had a significant direct or indirect impact on their ICT teaching behavior based on the modified UTAUT model, the correlational nature of the study limits our ability to make inferences regarding causality. Second, to obtain large-scale data and explore the relationship between variables, the data were based on teachers’ self-reports for convenience and efficiency, which may be affected by the social desirability bias [57]. Further studies might need to consider collecting multi-modal data, such as objectively presenting teachers’ ICT use behavior by observing their actual teaching behavior and in-depth understanding of teachers’ perception of ICT teaching through interviews, to supplement and verify the results based on large-scale surveys. Third, the current research results are based on the overall group of teachers in China. We would explore the applicability of the current model in different regions or colleges with different development levels and consider the different genders of teachers in further studies.

## Figures and Tables

**Figure 1 behavsci-13-00077-f001:**
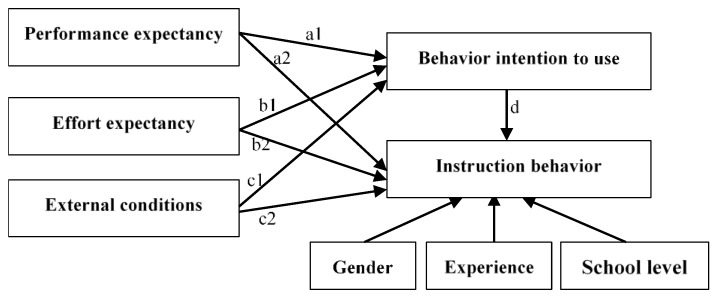
A hypothesized model of the influence of teachers’ performance expectancy, effort expectancy, and external conditions on their instruction behavior.

**Figure 2 behavsci-13-00077-f002:**
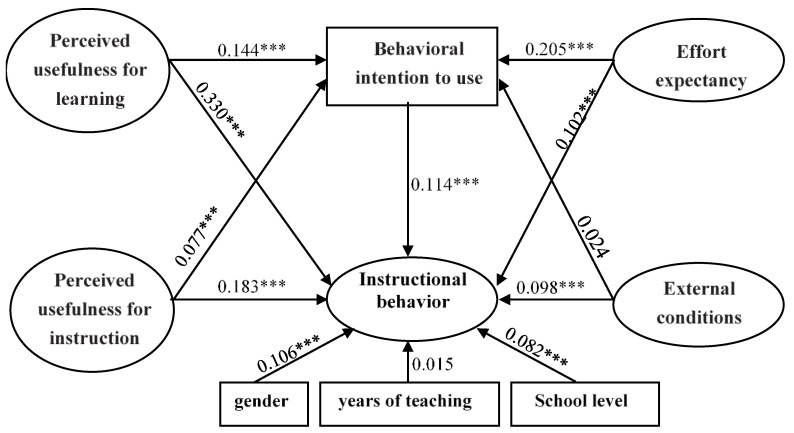
The resulting model of the relationship among teachers’ performance expectancy, effort expectancy, external conditions, behavioral intention, and their instructional behavior (N = 6087, χ^2^/df = 23.05, RMSEA = 0.060, CFI = 0.905, TLI = 0.900). *** means the correlation was significant at 0.001 or below (*p* < 0.001).

**Table 1 behavsci-13-00077-t001:** Distribution of teachers’ personal variables (N = 6087).

Personal Variables		N	%
**Gender**	Male	2194	36.0%
	Female	3893	64.0%
**Age**	25 or below	158	2.6%
	26–40	3241	53.2%
	41–55	2297	37.7%
	55 or high	391	6.4%
**Years of teaching experience**	Less than 1 year	367	6.0%
	1–3	691	11.4%
	4–6	634	10.4%
	7–18	2668	43.8%
	19–30	1256	20.6%
	31–40	471	7.7%
**Years of working experience**	0	2663	43.7%
	Less than 1 year	1551	25.5%
	1–3	854	14.0%
	4 or high	1019	16.7%
**Professional rank**	No rank	431	7.1%
	Primary	963	15.8%
	Secondary	2790	45.8%
	Senior	1903	31.3%
Highest degree	Bachelor’s	1479	24.3%
	Master’s	3888	63.9%
	Ph.D.	241	4.0%
	Others	479	7.9%

**Table 2 behavsci-13-00077-t002:** The Cronbach’s alpha reliability coefficients and CFA model fit of the predictive and outcome variables.

	Cronbach’s Alpha Reliability Coefficients	CFI	TLI	RMSEA	Factor Loadings
**Perceived usefulness for learning**	0.974	0.934	0.916	0.142	0.822–0.896
**Perceived usefulness for instruction**	0.946	0.943	0.905	0.188	0.785–0.891
**Effort expectancy**	0.920	0.936	0.900	0.150	0.444–0.956
**External conditions**	0.940	0.962	0.944	0.118	0.691–0.912
**Teachers’ instructional behavior**	0.936	0.929	0.910	0.102	0.418–0.862

**Table 3 behavsci-13-00077-t003:** The results of convergent (CR), discriminant validity (AVE), and correlation analysis.

	CR	AVE	1	2	3	4	5
**1. Perceived usefulness for learning**	0.972	0.757					
**2. Perceived usefulness for instruction**	0.944	0.739	0.630 ***				
**3. Effort expectancy**	0.914	0.582	0.298 ***	0.388 ***			
**4. External conditions**	0.940	0.664	0.481 ***	0.559 ***	0.372 ***		
**5. Behavioral intention to use**	-	-	0.252 ***	0.247 ***	0.310 ***	0.202 ***	
**6. Teachers’ instructional behavior**	0.936	0.537	0.536 ***	0.500 ***	0.406 ***	0.417 ***	0.296 ***

Note: *** means the correlation was significant at 0.001 or below (*p* < 0.001).

**Table 4 behavsci-13-00077-t004:** The mediating chain analysis of the relationship between predictive variables and teachers’ instructional behavior.

		Effect Size	*p*	Coverage
Perceived usefulness for learning-> Teachers’ instructional behavior	Indirect effect (d**e*)	0.016	<0.001	4.62%
	Direct effect	0.330	<0.001	95.38%
	Total effect	0.346		100%
Perceived usefulness for instruction -> Teachers’ instructional behavior	Indirect effect (d**e*)	0.009	<0.001	4.69%
	Direct effect	0.183	<0.001	95.31%
	Total effect	0.192		100%
Effort expectancy -> Teachers’ instructional behavior	Indirect effect (d**e*)	0.023	<0.001	18.40%
	Direct effect	0.102	<0.001	81.60%
	Total effect	0.125		100%
External conditions -> Teachers’ instructional behavior	Indirect effect (d**e*)	0.003	0.144	2.97%
	Direct effect	0.098	<0.001	97.03%
	Total effect	0.101		100%

## Data Availability

The data in this study can be provided upon request by sending an e-mail to the corresponding author.

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
