# Peer review of "What Affects Vocational Teachers’ Acceptance and Use of ICT in Teaching? A Large-Scale Survey of Higher Vocational College Teachers in China"

_behavsci, 2023, doi:10.3390/bs13010077_

Round 1

Reviewer 1 Report

I think it is a clear and rigorous article. Although not original, it applied the modified UTAUT model to an important sample of vocational teachers in China. The methodology followed is the usual one in this type of study and was well applied. The results are congruent with other studies that use this model. The authors also point out practical implications for the training of vocational teachers in China and the limitations of the study.

Author Response

We would like to thank “reviewer 1” for his/her review and comments. In light of the comments of “reviewer 1”, we have further checked and enhanced the language of the article.

Reviewer 2 Report

Title: What Affects Vocational Teachers' Acceptance and Use of ICT in Teaching? A Large-scale Survey of Higher Vocational College Teachers in China

Comments:

1.       The current study makes use of primary data derived from the survey of 6087 teachers from 219 vocational colleges in 28 provinces of China. It examined the factors affecting the vocational teacher’s acceptance and use of ICT in teaching. It proved that teachers’ internal psychological perception and external factors supported by institutions had a significant direct or indirect impact on their teaching behavior using ICT using the UTAUT model.

2.       The succinct discussion of the paper is truly commendable. In pithy words, the juxtaposing of concepts –  both internal and external conditions on the use of ICT in teaching – was commendably executed in a well-woven narrative.

3.       Scientific content is empirically sound and prudently tested. Interpretation of the data coheres with the statistical findings. The current findings are even combined with previous findings (references supplied by the writers) on the possible reasons for the correlation.

4.       The conclusion answers the research questions stated and the hypotheses, soundly tested and measured, carefully supported the statistical analysis of the data.

5.       Observed findings, though with stated limitations, were ingeniously and adequately extrapolated, transcending whatever barriers they encountered in obtaining research data.

6.       One article similar to this study may be included as an additional reference as it touches almost on the same topic with almost the same variables and almost the same practical implications: Mirzajani, H., Mahmud, R., Fauzi Mohd Ayub, A. and Wong, S.L. (2016), "Teachers’ acceptance of ICT and its integration in the classroom", Quality Assurance in Education, Vol. 24 No. 1, pp. 26-40. https://doi.org/10.1108/QAE-06-2014-0025

7.       Minor grammar lapses were observed particularly in lines 339-341 and in other parts of the manuscript. Please check line 59 as well.  

Author Response

Thank you very much to “Reviewer 2” for reviewing and commenting. Based on Reviewer 2's comments, we have made the following changes to the article.

(1) The language of the paper has been further checked and enhanced. For example, line 59 was revised from “In Addition, teachers’ psychological perceptions, such as cognition, attitudes, and motivation, also have important implications for boosting the use of ICT in VET teaching, as teaching is emotional labor” to “In Addition, teachers’ psychological perceptions, such as cognition, attitudes, and motivation, also have important implications for boosting the use of ICT in VET teaching”; line 339-341 was revised from “However, there were differences in the significance of these variables in the chain effect through teachers’ intention to use ICT, which was embodied in hypotheses H1, H2, H3, H4, and H5 were accepted, while hypothesis H6 was rejected.” to “However, there were differences in the significance of these variables in the chain effect through teachers’ intention to use ICT: hypotheses H1, H2, H3, H4, and H5 were accepted, while hypothesis H6 was rejected.”.

(2) We read the literature recommended by "Reviewer 2" and added it as a reference in the 'Conclusion and Implications' section of the article.